# *Ubinuclein 2* is essential for mouse development and functions in X chromosome inactivation

Asun Monfort[1¤], Giulio Di Minin[1], Sarah Sting[1], Charles Etienne Dumeau[1,2], Peter Scambler[3], Anton Wutz[1*]

**1** Institute of Molecular Health Sciences, Swiss Federal Institute of Technology, Zurich, Switzerland, **2** Friedrich Miescher Institute for Biomedical Research (FMI), Basel, Switzerland, **3** UCL Great Ormond Street Institute of Child Health, University College London, London, United Kingdom

¤ Current address: HBM Partners, Zug, Switzerland
* awutz@ethz.ch

## Abstract

The HIRA complex mediates deposition of histone H3.3 independent of replication. Its functions in gene regulation in mice remain to be fully understood. Here we analyze mutations of the HIRA complex genes *Ubn1* and *Ubn2*. We observe that *Ubn1* mutant mice of both sexes are viable and fertile. In contrast, mutation of *Ubn2* causes embryonic lethality with variable penetrance and skewed sex ratio in favor of males. Combined *Ubn1* and *Ubn2* mutations cause embryonic lethality with complete penetrance, variable developmental arrest before turning, and reduced recovery of female embryos. Consistent with a female specific function of the HIRA complex, reanalysis of the *Hira* mutation during embryogenesis reveals that previously observed severe and mild phenotypic classes correspond to female and male sex. Mechanistically, we show that mutations of *Ubn1*, *Ubn2*, and *Hira* in mouse embryonic stem cells affect the initiation of X inactivation. *Xist* mediated gene silencing is impaired to increasing extent by *Ubn1*, *Ubn2*, *Hira*, and combined *Ubn1* and *Ubn2* mutations. We identify a failure of establishing histone H3 tri-methyl lysine 27 over X-linked genes after induction of *Xist* expression as earliest molecular defect, whereas deacetylation of lysine 27 by *Xist* remains largely unaffected by the loss of Ubinucleins. Our study thereby identifies a switch from histone H3 acetyl to tri-methyl lysine 27 at the initiation of X inactivation that depends on HIRA complex function.

## Author summary

The genome of all eukaryotes is assembled into a nucleosomal structure, which is accomplished by deposition of histone proteins on the DNA. This process involves distinct mechanisms that are dependent or independent of DNA replication. The evolutionary conserved HIRA complex deposits histone H3.3 independent of DNA

**Data availability statement:** ChIP-Seq, RNA-Seq, ATACseq, and CUT&RUN datasets are available from the SRA repository of the National Center for Biotechnology Information under the following SRA accession numbers: ChIP-Seq PRJNA574616; RNA-Seq: PRJNA574610; ATACseq and CUT&RUN: PRJNA579971.

**Funding:** This work was supported by the Swiss National Science Foundation (SNF grants 31003A_152814/1 and 31003A_175643/1 to AW) and the ETH Zurich (grant ETH-38 16-1 to AW). The funders had no role in the study design, data collection and analysis, decision to publish, or preparation of the manuscript.

**Competing interests:** The authors have declared that no competing interests exist.

replication within active gene loci and regulatory regions. Its functions in gene regulation remain to be fully understood in animal development. By analyzing mutations in HIRA complex genes in mice, we discover a female specific function of the HIRA complex in embryogenesis. HIRA complex function is required for establishing gene silencing on the inactive X chromosome and dosage compensation in mice. We identify a switch from acetyl to tri-methyl modified histone H3 lysine 27 at X-linked genes during the initiation of X chromosome inactivation as molecular event that is dependent on HIRA complex function. Our study provides a characterization of the function of HIRA complex genes in mouse development and identifies a novel function in the establishment of facultative heterochromatin on the inactive X chromosome.

## Introduction

Histone H3.3 is implicated in different processes including transcription [1], DNA repair [2], and genome stability [3]. Additionally, mutations in histone H3.3 occur in neurodevelopmental tumors [reviewed in 4]. In contrast to canonical histones H3.1 and H3.2, histone H3.3 deposition into chromatin is not restricted to S-phase. Histone regulator A (HIRA) and death domain associated protein (DAXX) complexes mediate histone H3.3 deposition by distinct mechanisms [reviewed in 5]. The latter forms a complex with alpha-thalassemia/mental retardation, X-linked (ATRX) integrating histone H3.3 into heterochromatin at telomeres, centromeres, and specific genomic elements which are associated with histone H3 tri-methyl lysine 9 (H3K9me3) [6,7]. Mutation of *Atrx* causes embryonic lethality in mice due to genome instability and trophoblast defects [8]. HIRA mediates histone H3.3 deposition in the transcription unit of active genes and at regulatory elements including superenhancers [9–11]. In addition, *Hira* is essential for reprogramming the zygotic genome after fertilization [12,13]. Mutation of *Hira* causes embryonic lethality shortly after implantation with mesodermal and patterning defects [14]. Surprisingly, mutation of *Hira* in embryonic stem cells (ESCs) has only minor effects on transcription [15] but *Hira* becomes essential when cells differentiate. Furthermore, HIRA and histone H3.3 are enriched at bivalent promoters and facilitate the establishment of histone H3 tri-methyl lysine 27 (H3K27me3) by Polycomb repressive complex 2 (PRC2) [16]. HIRA associates with the N-terminal NHRD domain of UBN1 through its WD repeats [17]. UBN1 binds histone H3.3 through an N-terminal Hpc2-related domain (HRD) [10] and has specificity for 3 amino acids in the core of histone H3.3 that differ from canonical histones H3.1 and H3.2 [4]. In mice, two genes, *Ubn1* and *Ubn2*, encode proteins with homology to the Drosophila *Yemanuclein* and yeast *Hpc2* genes [4]. Both share extensive similarity and have been shown to associate with HIRA [18,19], but their functions remain to be determined in mice.

We have previously identified *Hira* and *Ubn2* as candidate genes for silencing factors in X chromosome inactivation (XCI) [20]. XCI is the mechanism for dosage compensation between males and females in mammals [reviewed in 21]. The non-coding *Xist* RNA is transcribed from and associates specifically with the inactive

X chromosome (Xi) leading to silencing of X-linked genes [22,23]. Spreading of *Xist* over the Xi forms a repressive compartment that is depleted of RNA polymerase II, splicing and transcription factors, as well as acetylated histone H3 and H4 [24]. Subsequently, PRC1 and PRC2 are recruited and establish mono-ubiquitination of histone H2A lysine 119 (H2AK119ub) and H3K27me3, respectively [25,26]. Recruitment of Polycomb complexes is facilitated by *Xist* repeats B and C, and a pathway involving heterogeneous ribonucleoprotein K (hnRNPK) [25,27–29]. Notably, the initial repressive compartment forms over regions that are devoid of genes and genes only associate with the Xist domain upon silencing [24]. Binding of the split ends protein SPEN to *Xist* repeat A is required for X-linked gene silencing by an HDAC3 dependent mechanism [20,27,30,31]. In somatic cells, the Xi forms a stable heterochromatic structure and gene repression becomes independent of *Xist* [reviewed in 21].

We previously used an inducible *Xist* expression system in mouse haploid ESCs for screening of silencing factors in XCI [20]. Induction of *Xist* led to rapid silencing of the single X chromosome and caused cell death. Strong selection for *Ubn2* and *Hira* mutations in surviving cells after *Xist* induction indicated a potential role in XCI. Specifically, we hypothesize that *Hira* and *Ubn2* might function in Polycomb recruitment and gene silencing. Here, we engineer *Ubn1* and *Ubn2* mutations in mice and ESCs to characterize their role in embryonic development and XCI.

## Results

### *Ubn1* mutant mice are viable and fertile

*Ubn1* and *Ubn2* are the mammalian ortholog and paralog of the *Drosophila Yemanuclein* and yeast *Hpc2* genes, respectively [1,4,32] (Fig 1A and 1B). In mouse zygotes, RNA interference mediated depletion of *Ubn1* has been reported to abrogate preimplantation development [33]. We previously engineered an approximately 10kb deletion of *Ubn1* exons 1–7 encoding the N-terminal NHRD and HRD domains (also encompassing the HUN domain), which are required for interaction with HIRA and histone H3.3 [10,17,32] (Fig 1C) in mice. For this we performed zygote electroporation of a pair of Cas12a/Cpf1 nuclease gRNA complexes and obtained several founder mice in the C57BL/6 background [34]. From crosses we obtained homozygous *Ubn1* mutant offspring of both sexes that appeared healthy and fertile. We subsequently established a homozygous *Ubn1* mutant mouse colony and confirmed the absence of UBN1 protein in liver nuclear extracts by Western analysis (Fig 1D). Over the past 5 years no overt phenotypic alteration was observed and average litter sizes of 5.31 +/- 2.3 and 5.93 +/- 2.2 were calculated for *Ubn1*[-/-] and control wildtype C57BL/6 mice, respectively (n > 160 litters) suggesting comparable breeding success. Both *Ubn1*[-/-] and wildtype colonies showed a balanced sex ratio. We conclude that *Ubn1* is not essential in mice.

### Embryonic lethality of the *Ubn2* mutation with incomplete penetrance and male biased survival

The absence of a developmental phenotype in *Ubn1* mutant mice indicated possible compensation by *Ubn2*. We therefore engineered a 17kb deletion of *Ubn2* exons 3–7 encoding the N-terminal HUN domain [32] that is predicted to introduce a frameshift (Fig 2A). We obtained several founders with deletions and inversions in *Ubn2* from pronuclear microinjection of a pair of Cas9 nuclease-gRNA complexes into C57BL/6 zygotes. Crosses of heterozygous *Ubn2*[+/-] mice yielded several homozygous *Ubn2*[-/-] mutant males at submendelian frequency that were healthy and fertile (Fig 2B). However, initially no females were found at weaning. A female specific lethality was further substantiated by crossing homozygous *Ubn2*[-/-] males with heterozygous *Ubn2*[+/-] females (Fig 2B). Taken together, all crosses yielded only 7 *Ubn2*[-/-] females against a total of 48 *Ubn2*[-/-] males (S1 Table). Therefore, homozygous *Ubn2* mutant females in the C57BL/6 background were rarely obtained and did not produce offspring when mated with either mutant or wild type males. To determine at which developmental stage *Ubn2*[-/-] embryos were lost, we analyzed embryos at embryonic days (E) 10.5, E11.5, and E12.5. We recovered mutant embryos of both sexes and observed a large variability of embryonic delay (Figs 2C, and S1). Some male homozygous mutant embryos appeared comparable to control heterozygous embryos consistent with survival of *Ubn2*[-/-] males at submendelian frequency. Homozygous *Ubn2*[-/-] mutant female embryos could be recovered at all stages

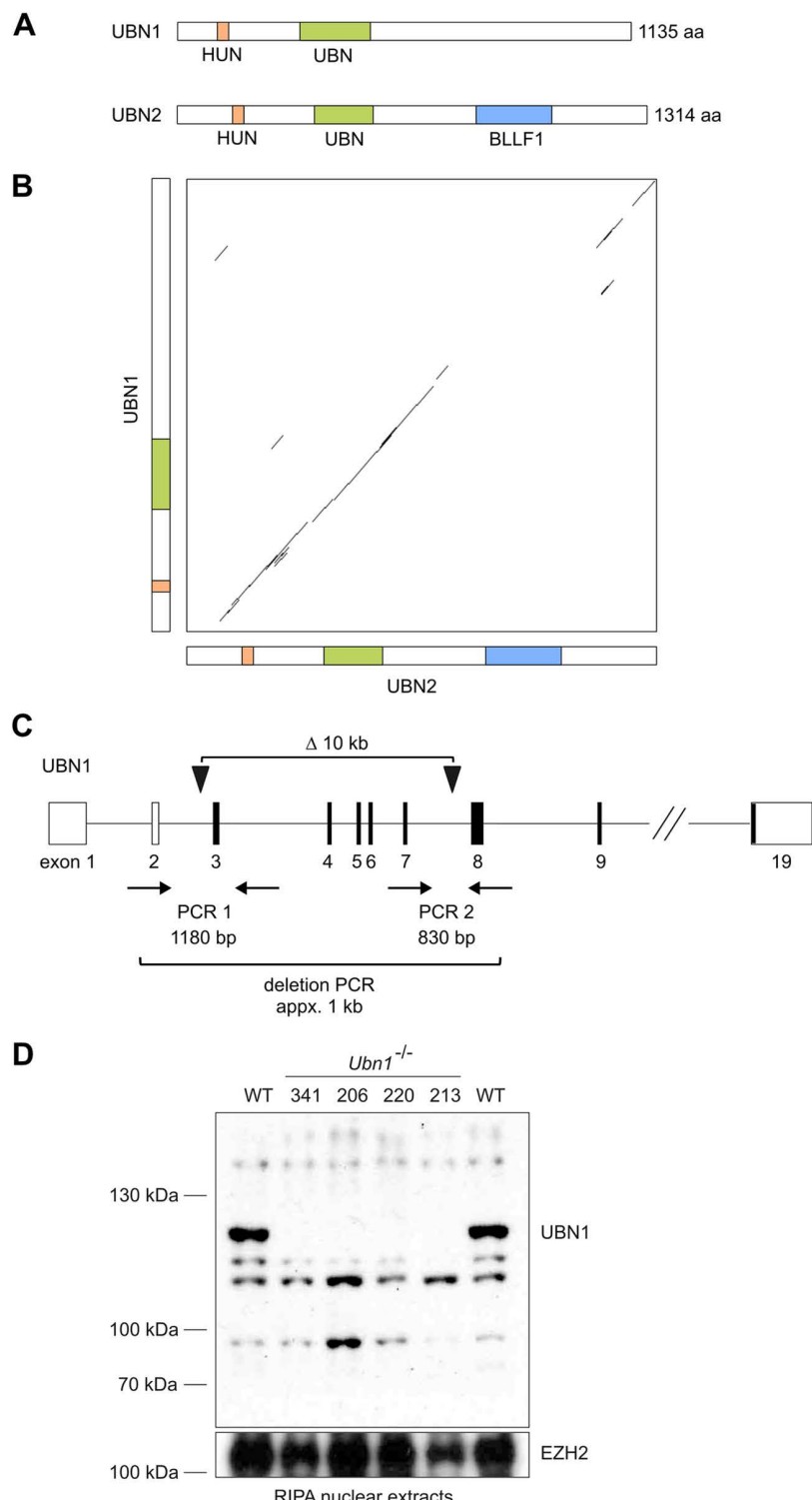

**Fig 1. *Ubn1* mutation in mice. (A)** Scheme showing the domain structure, and **(B)** Dot-plot showing protein similarity (Dotmatcher, windowsize = 30, threshold = 40) of the mouse UBN1 and UBN2 proteins. **(C)** Scheme showing the strategy for engineering a deletion in *Ubn1* in mice. **(D)** Western analysis of UBN1 protein in liver nuclear extracts. EZH2 was used to control for loading.

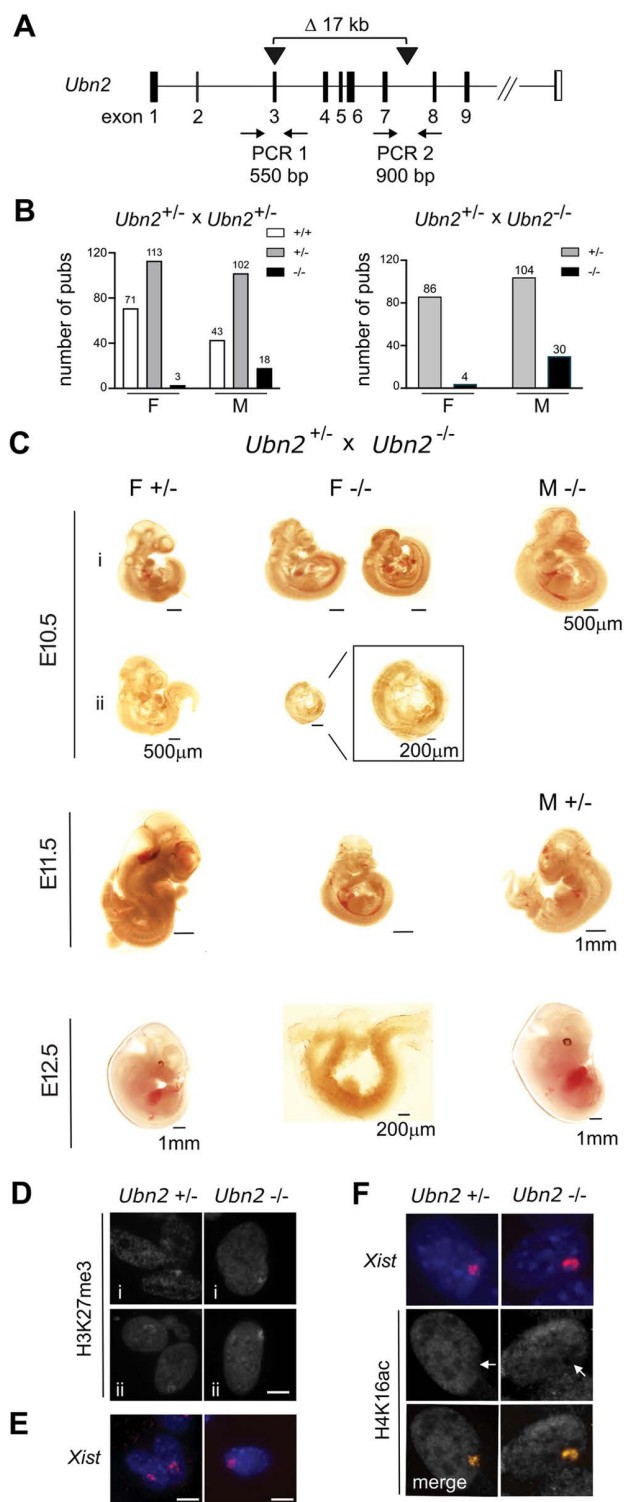

**Fig 2. *Ubn2* mutation causes a female specific lethality with high penetrance in C57BL/6 mice. (A)** Scheme showing the strategy for engineering a deletion in *Ubn2* in mice. **(B)** Male (M) and female (F) offspring from crosses of mice as indicated. Numbers above columns represent the number of mice. **(C)** Female specific phenotype of *Ubn2⁻/⁻* embryos at E10.5, 11.5, and 12.5. i) and ii) represent two independent litters. **(D)** H3K27me3 immunofluorescence, **(E)** *Xist* RNA FISH, and **(F)** Combined *Xist* RNA FISH and H4ac immunofluorescence of E11.5 *Ubn2⁻/⁻* mouse embryonic fibroblasts (MEFs). i, and ii represent two independent cultures. Scale bar, 5 μm.

but showed a developmental delay with frequent arrest before E10.5 (Fig 2C). We established fibroblasts from two E11.5 female *Ubn2*⁻/⁻ embryos. Immunofluorescence and RNA FISH analysis revealed H3K27me3 foci and *Xist* clusters comparable to controls suggesting that XCI had been initiated (Fig 2D and 2E). Furthermore, H4K16ac was depleted from the *Xist* domains indicating that *Xist* formed a repressive compartment (Fig 2F). We conclude that the *Ubn2* mutation causes embryonic lethality with incomplete penetrance and biased survival of males in the C57BL/6 background.

## Combined *Ubn1* and *Ubn2* mutations cause embryonic lethality with complete penetrance

To investigate if *Ubn1* mediated the survival of *Ubn2*⁻/⁻ mice, we introduced the *Ubn2* mutation into the *Ubn1* homozygous mutant C57BL/6 background by crossing. We obtained mice with heterozygous *Ubn2* and homozygous *Ubn1* mutations of both sexes. However, we did not obtain double homozygous mutant *Ubn1*⁻/⁻ *Ubn2*⁻/⁻ mice suggesting embryonic lethality. We next analyzed embryos from crosses of *Ubn1*⁻/⁻ *Ubn2*⁺/⁻ mice at E10.5. We recovered several homozygous double-mutant *Ubn1*⁻/⁻ *Ubn2*⁻/⁻ embryos that showed an embryonic delay and arrest before turning (Fig 3A). We performed Chi-Square tests and observed a high deviation from expectations for double homozygous embryos that, however, did not reach statistical significance (S2 Table). We also noted an elevated frequency of *Ubn2* heterozygous and *Ubn1* homozygous mutants that might indicate potential maternal contamination. All genotypes approach Mendelian frequencies if we consider that maternal contamination occurred at some poorly developed *Ubn1*⁻/⁻ *Ubn2*⁻/⁻ implantation sites. None of the *Ubn1*⁻/⁻ *Ubn2*⁻/⁻ embryos had initiated turning demonstrating a severe developmental delay. The high variability of

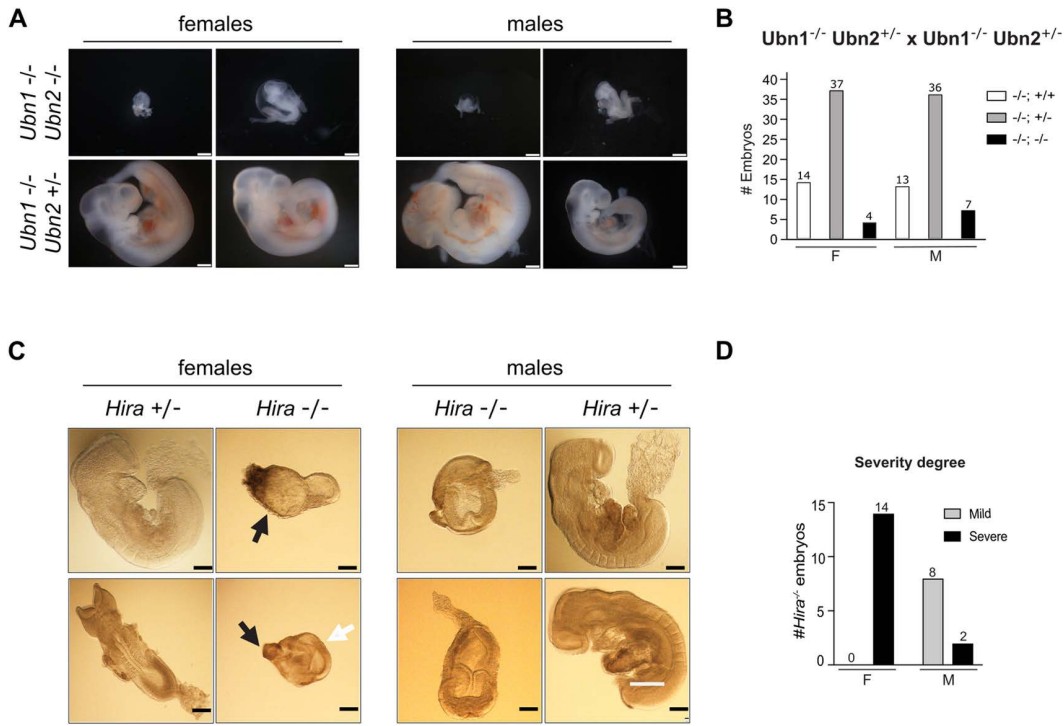

**Fig 3. Early embryonic lethality of the *Ubn1* and *Ubn2* double and *Hira* mutations in mice. (A)** Representative images of male and female *Ubn1*⁻/⁻ *Ubn2*⁻/⁻ and control *Ubn1*⁻/⁻ *Ubn2*⁺/⁻ E10.5 embryos. Scale bar, 500 µm. **(B)** Genotype distribution of embryos obtained from crossing *Ubn1*⁻/⁻ *Ubn2*⁺/⁻ mice. **(C)** Images of male (M) and female (F) *Hira*⁻/⁻ embryos at E8.5. Embryonic (white arrows) and extraembryonic tissues (black arrows) are indicated. Scale bar, 200 µm. **(D)** Phenotypic classes, mild and severe, of male (M) and female (F) *Hira*⁻/⁻ E8.5 embryos. Number of embryos is indicated on top of columns.

embryonic delay among *Ubn1*[-/-] *Ubn2*[-/-] E10.5 embryos was striking with a 2-fold higher recovery of *Ubn1*[-/-] *Ubn2*[-/-] male compared to female embryos (Fig 3B).

**Two phenotypic classes of the *Hira* mutation correlate with sex of the embryos**

Consistent with our observation in *Ubn1*[-/-] *Ubn2*[-/-] mice, embryonic lethality after gastrulation has previously been observed for the *Hira* mutation in inbred C57BL/6 mice. In the outbred CD1 background embryos could be recovered until E10.5, whereby two classes of phenotypes were reported, namely a mildly affected and a severely affected class [14]. Reduced recovery of female *Ubn2*[-/-] mice and *Ubn1*[-/-] *Ubn2*[-/-] embryos suggested the possibility that the two classes of the *Hira*[-/-] embryos were also correlated with sex. We therefore established crosses of heterozygous *Hira* mutant mice in the CD1 background. Among 146 embryos (73 females and males) were 14 and 10 *Hira*[-/-] females and males, respectively. All *Hira*[-/-] embryos were markedly smaller than wildtype or heterozygous littermates (Fig 3C). Female *Hira*[-/-] embryos showed a more severe developmental delay as judged by size and earlier arrest than male *Hira*[-/-] embryos (Fig 3D). When assigned into two phenotypic classes, all female embryos clearly associated with the severe phenotype. In contrast, the majority of male homozygous mutant embryos developed further and displayed milder phenotypes. *Hira*[-/-] embryos of both classes arrested before turning consistent with previous observations [14]. We conclude that sex of the embryo correlates with two phenotypic classes of the *Hira* mutation in the CD1 background.

***Hira* and *Ubn2* complexes contribute to *Xist* function in mouse ESCs**

Our finding that mutations in *Ubn2* and *Hira* affected female more than male development were consistent with previous identification of both genes in our screen for silencing factors in XCI [20]. To study their role in XCI further, we engineered mutations in ESCs that contain an inducible *Xist* expression system. A tetracycline-inducible promoter (tetOP) was inserted at the *Xist* transcription start site and a tetracycline-regulated transactivator targeted into the ubiquitously expressed ROSA 26 locus on chromosome 6 [20] (S2A Fig). Induction of *Xist* with doxycycline causes cell death due to X-linked gene silencing, which provides a readout for *Xist* function. Initially, these cells were derived as haploid ESCs [20], but became diploid over time and, hence, possess two X chromosomes from which *Xist* can be induced. We engineered *Ubn2* mutations in two independent ESC lines, HATX3 and HATX8 [20], using Cas9 nucleases with either a single gRNA to generate frame shift mutations in exon 3, or a pair of gRNAs to make a deletion from exon 3 to intron 7 (Fig 4A). We obtained several Δ*Ubn2* ESC lines that proliferated normally and maintained a typical colony morphology suggesting that self-renewal of ESCs was not impaired. UBN2 protein was undetectable by Western analysis (Figs 4B and S2B–S2D). To assess the effect of *Ubn2* mutations on *Xist* function, we measured the survival of ESCs after *Xist* induction using a clonal assay, where the number and size of colonies in the presence relative to the absence of doxycycline was recorded (S2E Fig). In the parental ESCs *Xist* induction drastically reduced colony size and number, whereas three independent Δ*Ubn2* ESC clones showed greater survival after *Xist* induction than the parental WT ESCs (Figs 4C, S2F and S2G). Importantly, complementation of the *Ubn2* mutation by stable integration of a piggyBac vector expressing a *Ubn2* cDNA transgene restored *Xist* function (Figs 4B, 4C, S2F and S2H). Similarly, we engineered mutations in *Hira* (Figs 4D and S2I and S2J) and confirmed the absence of HIRA protein in Δ*Hira* ESCs by Western analysis (Fig 4E). Independent Δ*Hira* ESC clones showed increased cell survival after *Xist* induction compared to wildtype controls (Figs 4F and S2K). To confirm formation of complexes containing UBN2 and HIRA in our ESC system, we introduced a hemagglutinin-tag (HA) into a C-terminal position of the endogenous *Ubn2* gene locus using Cas9 nuclease mediated homology directed repair. HA-tagged UBN2 protein in *Ubn2*-HA ESCs was detected by Western analysis (Fig 4G). Western analysis of HA-immunoprecipitates of nuclear extracts of *Ubn2-HA* ESCs showed that HIRA protein was present confirming that HIRA interacts with UBN2 (Fig 4H). To obtain additional support, we performed preliminary characterization of this complex by comparative proteomics of SILAC-labelled *Ubn2-HA* ESCs and control untagged ESCs. We observed enrichment of HIRA, UBN1, and CABIN1 in UBN2-HA over

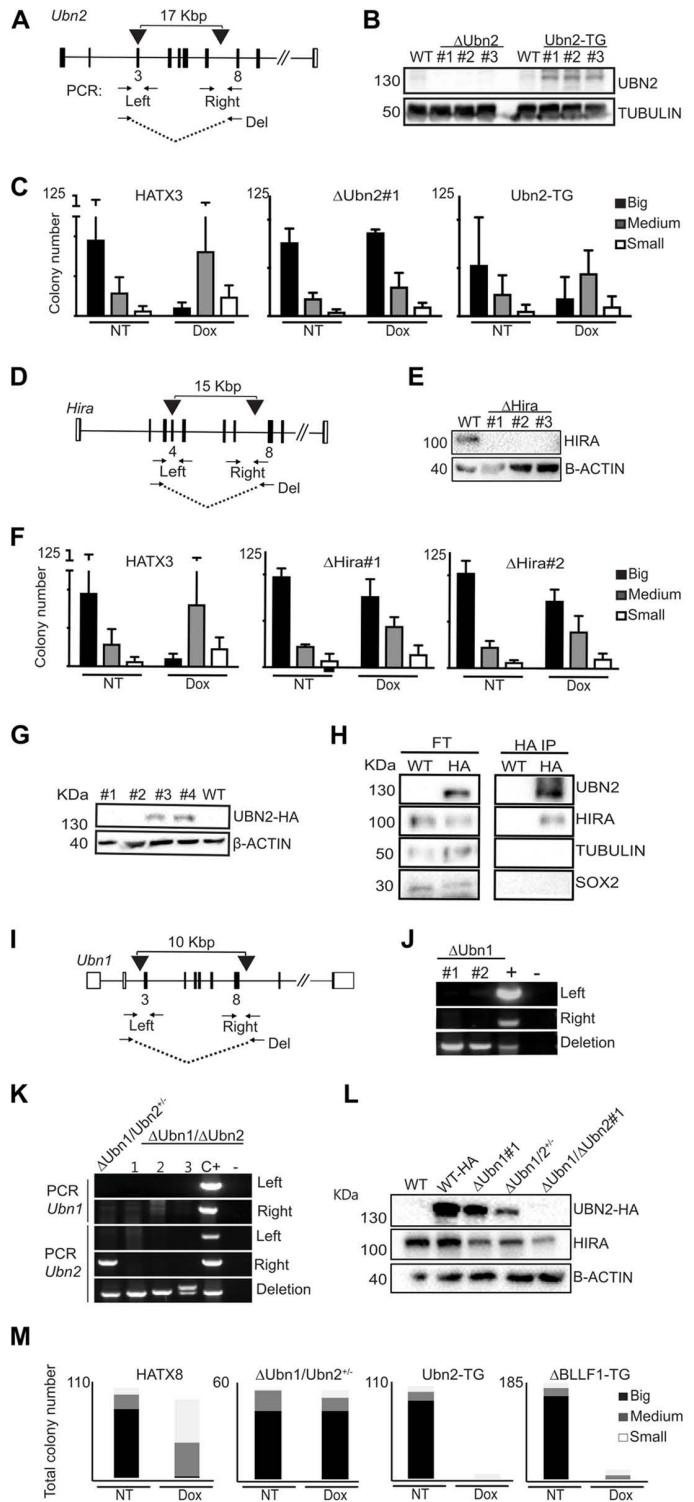

**Fig 4. *Ubn2* and *Hira* are required for XCI. (A)** Scheme showing the positions of gRNAs (black arrow heads) and PCR primers (arrows) for engineering deletions in Δ*Ubn2* clones *#1 and #2,* and a frame-shift in exon 3 in Δ*Ubn2#3*. **(B)** Western analysis of UBN2 protein in control WT, Δ*Ubn2*, and transgenically (TG) rescued Ubn2-TG ESCs is shown. TUBULIN used as loading control. **(C)** Representative *Xist* survival assay for control HATX3, Δ*Ubn2*, and rescued Ubn2-TG ESCs as indicated. For each cell line colony numbers binned in 3 size classes are shown without (NT) or with (Dox) *Xist* induction (n = 3). **(D)** Strategy for disrupting *Hira* as in **(A)**. Deletions were introduced into Δ*Hira* clones *#1 and #2,* and a frame-shift in exon 4 in

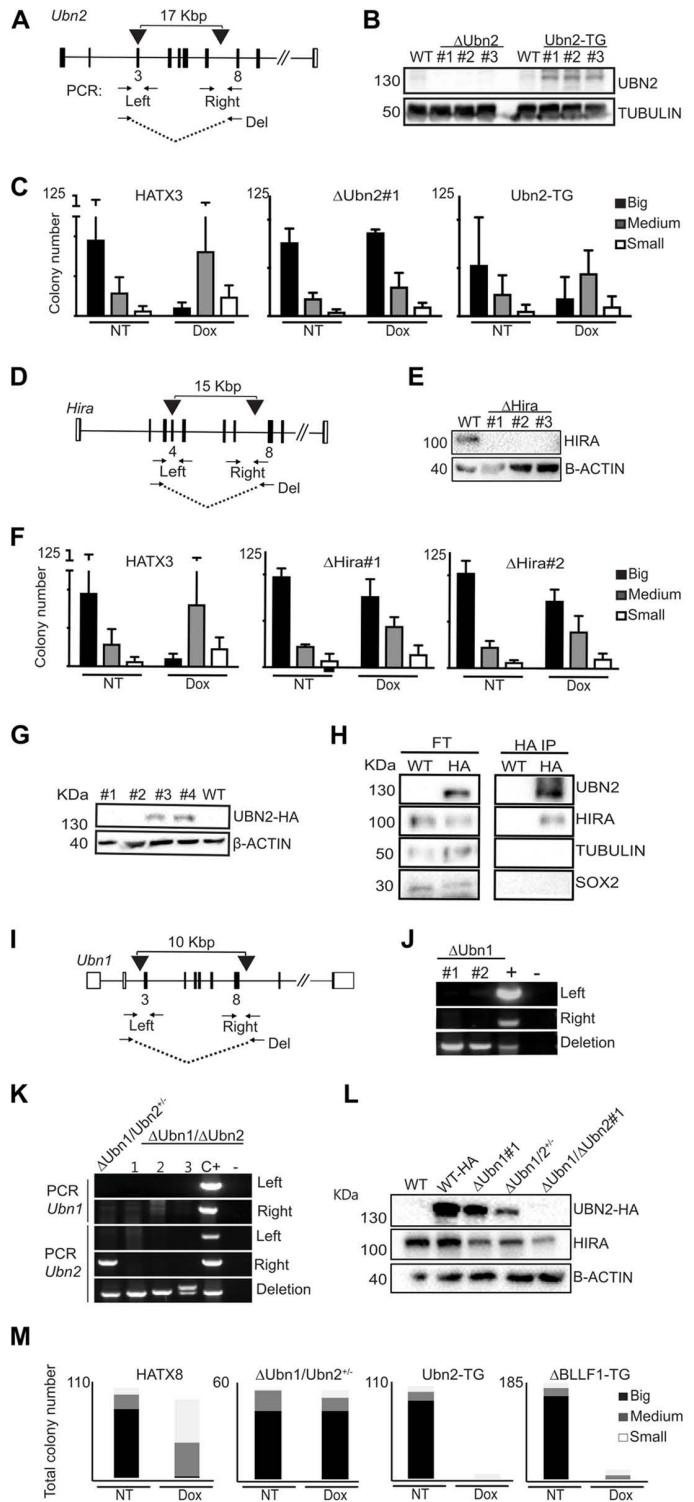

ΔHira#3. **(E)** Western analysis of HIRA protein in control WT, and ΔHira ESCs. β-ACTIN used as loading control. **(F)** Representative *Xist* survival assay for control HATX3, ΔHira#1 and ΔHira#2 ESC clones without (NT) or with (Dox) *Xist* induction (n = 3) is shown as in **(C)**. **(G)** Western analysis showing UBN2-HA protein expression in *Ubn2*-HA ESC clones #3 and #4 using HA antisera. β-ACTIN used as loading control. **(H)** Western analysis detecting HIRA in HA-immunoprecipitates from nuclear extracts of *Ubn2*-HA ESCs using antisera as indicated. FT; flow through. **(I)** Strategy for disrupting *Ubn1* in Ubn2-HA ESCs as in **(A)**. **(J)** Genomic PCR identifying deletions in ΔUbn1 ESC clones #1 and #2. +; positive control heterozygous for the ΔUbn1 deletion. **(K)** Genomic PCR showing deletions in four independent ΔUbn1/ΔUbn2 double mutant ESC clones. A band is observed for the right flanking PCR in ΔUbn1/ΔUbn2⁺ᐟ⁻ ESCs indicating a partial deletion. C+; positive control heterozygous for the ΔUbn1 and ΔUbn2 deletions. **(L)** Western analysis of UBN2-HA and HIRA in WT, ΔUbn1, and ΔUbn1/ΔUbn2 ESCs. ΔUbn1/ΔUbn2⁺ᐟ⁻ ESCs show a reduced expression of truncated UBN2-HA and appear to be functionally comparable to ΔUbn1/ΔUbn2 ESC clones. β-ACTIN used as loading control. **(M)** Representative *Xist* survival assay for ΔUbn1/ΔUbn2⁺ᐟ⁻ ESCs rescued with a WT UBN2 and a ΔBLLF1-UBN2 cDNA transgene.

control precipitates (S3 Table), which is further consistent with a recent report [10]. Taken together our experiments showed that UBN2 and HIRA form complexes in ESCs and contribute to *Xist* function.

To further investigate if *Ubn1* also contributes to *Xist* function, we engineered an *Ubn1* deletion in an *Ubn2*-HA ESC line that was homozygous for the HA-tagged *Ubn2* allele (Fig 4G and 4I). We also generated a *Ubn1* and *Ubn2* double muta-tion by subsequently introducing a deletion into *Ubn2*. We obtained several *Ubn2*-HA ΔUbn1 mutant ESC clones (Fig 4J) and recovered one *Ubn2*-HA ΔUbn1/Ubn2⁺ᐟ⁻ (with a heterozygous *Ubn2* mutation, where one allele maintains sequences on the right flank, Fig 4K) and three homozygous *Ubn2*-HA ΔUbn1/ΔUbn2 ESC clones (Fig 4K). Western analysis con-firmed the reduction and absence of the *Ubn2*-HA protein in *Ubn2*-HA ΔUbn1/Ubn2⁺ᐟ⁻ and *Ubn2*-HA ΔUbn1/ΔUbn2 ESC clones, respectively (Fig 4L). *Ubn2*-HA ΔUbn1 ESCs showed increased survival after *Xist* induction, when compared to the parental WT cells (S3A Fig). *Ubn2*-HA ΔUbn1/ΔUbn2 double mutant ESCs showed higher survival after *Xist* induction than ΔUbn1 cells and comparable to the survival of ΔUbn2 ESCs (S3A, S3B Fig). Thus, albeit both Ubinucleins contribute to *Xist* function, *Ubn2* has a predominant role. UBN2 possesses a BLLF1 domain of unknown function that is absent in UBN1 (Figs 1A, 1B, and S3C). To investigate a role of the BLLF1 domain for *Xist* function, we introduced piggyBac vec-tors for stable expression of WT and ΔBLLF1 *Ubn2* cDNAs into ΔUbn2 and *Ubn2*-HA ΔUbn1/Ubn2⁺ᐟ⁻ ESCs. Both trans-genes restored *Xist* function with comparable efficiency suggesting that the BLLF1 domain is not essential for *Xist* function (Figs 4M, and S3B). Transgenic rescue of *Xist* function confirmed that the effects on *Xist* were specific and caused by the *Ubn2* mutation.

### *Xist* mediated X-linked gene repression is impaired by *Hira* and *Ubn2* mutations

We next investigated the effect of our mutations on the transcriptome of ESCs and the repression of X-linked genes by *Xist*. For this, we performed RNA-Seq for three biological replicates of WT, ΔUbn1, ΔUbn2, ΔUbn1/ΔUbn2, and ΔHira ESCs either without or after 48 hours of *Xist* induction. RNA-Seq profiles for the different clones confirmed the expected absence of transcripts overlapping the deleted exons (S4A and S4B Fig). Analysis of gene expression differences to WT controls without *Xist* induction using DESeq2 [35] revealed that only 5 and 13 genes out of 15329 with nonzero total read count were significantly differentially expressed (adj.p-value < 0.05) in *Ubn1* and *Ubn2* mutant ESCs, respectively (S4 Table). In ΔUbn1/ΔUbn2 and ΔHira ESCs, 1745 and 703 genes were significantly differentially regulated, respectively (S4C Fig, and S4 Table). Of these, 222 and 159 genes were 2-fold upregulated and downregulated in *Hira* mutant ESCs, respectively. 533 and 449 genes were 2-fold upregulated or downregulated in ΔUbn1/ΔUbn2 ESCs. Notably, there was little overlap of differentially expressed genes between ESCs carrying different mutations (Fig 5A). Some of the downregu-lation might be explained by loss of the second X chromosome in ΔUbn1/ΔUbn2 ESCs (S3D Fig), but also among upreg-ulated genes only a minority overlaps between ΔHira and ΔUbn1/ΔUbn2 ESCs, which suggests biological differences. Known regulators of pluripotency remained unaffected by any of the mutations (S4 Table), which was further consistent with robust ESC self-renewal. *Hira*, *Ubn1*, and *Ubn2* transcripts remained unaffected by any mutation in the respective other genes indicating that transcriptional compensatory upregulation does not occur in ESCs (S4 Table). In contrast,

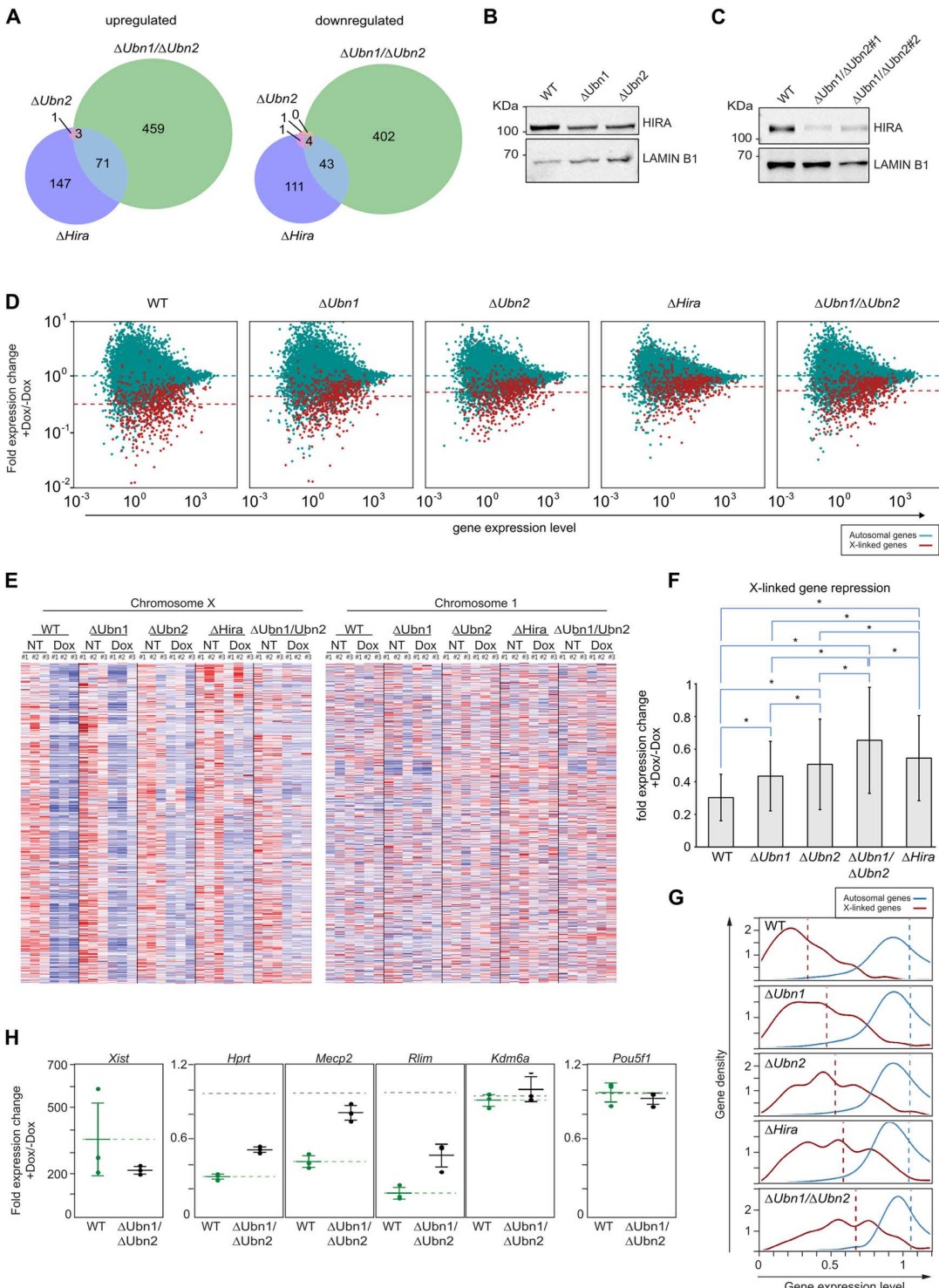

**Fig 5. Ubinucleins and *Hira* are required for efficient gene repression by *Xist*.** RNA expression analysis of WT, Δ*Ubn1*, Δ*Ubn2*, Δ*Hira*, and Δ*Ubn1*/Δ*Ubn2* ESCs. **(A)** Venn diagrams showing number of 2-fold or more up- (left) and down-regulated (right) genes in biological triplicates of Δ*Ubn2*, Δ*Hira* and Δ*Ubn1*/ Δ*Ubn2* relative to WT ESCs with an adjusted p-value cutoff of 0.05. **(B)** Western analysis of HIRA in Δ*Ubn1* and Δ*Ubn2* mice. **(C)** Western analysis of HIRA in Δ*Ubn1*/ Δ*Ubn2* cells. Lamin B1 was used as loading control. **(D)** Fold expression change of autosomal (green) and X-linked genes (red) after 48h of Dox treatment of ESCs with indicated genotypes versus their expression level in control condition. Dotted lines indicate the

median fold change of all autosomal (green) and X-linked genes (red). **(E)** Heatmap of normalized and z-transformed expression of genes in order of chromosomal positon is shown for ESCs of genotypes indicated, without (NT) and after 48h of *Xist* induction (Dox). Chromosome 1 is shown as auto-somal control. Red and blue color represent lower and higher expression from mean expression of the gene, respectively. White color indicates mean expression. **(F)** Bars graph of fold change expression of X-linked genes after 48h of *Xist* induction relative to uninduced controls. Genes with a fold-change of less than 0.6 in WT were selected and mean and standard deviation (errorbars) of fold change are shown for all genotypes after induction. *; p-value < 0.001 from a paired 2-sided t-test between groups at end of blue lines. **(G)** Efficiency of X-linked gene repression by *Xist*: Histograms (bins 0.1) show the distribution of gene numbers with indicated fold expression change after 48h of *Xist* induction (Dox) in combined replicates of WT and mutant ESCs. Autosomal (blue) and X-linked genes (red) are shown separately and their median fold expression change is indicated by vertical dotted lines. **(H)** Fold expression change for the X-linked *Xist*, *Hprt*, *Mecp2*, *Rlim*, and the escape genes *Kdm6a*, after 48h of treatment with Dox in WT and Δ*Ubn1*/ Δ*Ubn2* ESCs. *Pou5f1* is showed as an autosomal control. Dots represent replicates, mean and standard deviation are indicated by wide and narrow horizontal bars, respectively.

*Ubn1* and *Ubn2* mutations reduced HIRA protein level in mice (Fig 5B), and in Δ*Ubn1*/Δ*Ubn2* ESCs a drastic reduction of HIRA protein was evident by Western analysis (Fig 5C). This observation is consistent with previous reports of interdependence of HIRA complex proteins [2,10].

Next, we studied the effect of *Xist* induction on X-linked gene expression in the different mutant ESCs. In RNA-Seq datasets from ESCs that were treated with Dox for 48h, *Xist* was strongly up-regulated. Conversely, we observed a reduction in X-linked gene expression (Figs 5D, 5E, and S4D, S4E). Although a repression of X-linked genes was observed in all mutant ESCs, the strength of repression was markedly reduced compared to WT ESCs (Fig 5D–5G). In addition, we noted that *Xist* function was increasingly impaired along the genotype order Δ*Ubn1*, Δ*Ubn2*, Δ*Hira*, and Δ*Ubn1/Ubn2*. Loss of silencing efficiency of *Xist* was observed in a shift in the average fold expression change obtained over the three independent replicates for every mutation (Figs 5D, 5F, 5G and S4D) as well as when X-linked gene expression was represented in a heatmap for genes along the X-chromosome (Fig 5E). The observation that the *Hira* mutation and the *Ubn1* and *Ubn2* double mutation had the strongest impact on gene repression by *Xist* suggests redundant roles of *Ubn1* and *Ubn2* in ESCs (Fig 5D–5G). The strength of repression by *Xist* showed some variability over the X chromosome but was not correlated with the distance from the *Xist* locus (S4E Fig). Genes that escape XCI and autosomal genes were not affected by *Xist* induction (Fig 5E, 5H). Our results show that *Ubn2* and *Hira* are required for efficient *Xist*-mediated gene silencing, whilst mutation of *Ubn1* affected *Xist* only weakly and in combination with mutation of *Ubn2*.

### *Xist* forms a repressive compartment in the absence of *Ubn1* and *Ubn2*

Next, we investigated which steps of XCI depend on Ubinucleins. We performed RNA-FISH to examine *Xist* expression in WT, and Δ*Ubn1*/Δ*Ubn2* mutant ESCs (Fig 6). *Xist* clusters were observed after induction with doxycycline and their shape and size were comparable suggesting that *Xist* localization was not abrogated by the loss of Ubinucleins (Fig 6A). A slight decrease in the number of cells with Xist clusters was observed in Δ*Ubn1*/Δ*Ubn2* mutant compared to control ESCs (Fig 6B). To test if HIRA or UBN2 were enriched on the X chromosome after *Xist* induction, we performed immunofluorescence. We observed a diffusely punctate nuclear pattern for HIRA without a focal enrichment over the *Xist* domain (S5A Fig). Similarly, we did not detect a focus of UBN2-HA using an HA antiserum after induction of *Xist* in UBN2-HA ESCs (S5A Fig). Taken together our observations indicate that *Xist* does not recruit the HIRA complex suggesting that it might act locally on chromatin where it is already bound as has been previously suggested for hnRNPK, and HDAC3 [31].

We next investigated if *Xist* could form a repressive compartment. For this we performed combined immunofluorescence and *Xist* RNA FISH experiments using antibodies specific for H3K27me3 and H2AK119ub1, as well as JARID2 and EZH2 (Figs 6C, 6E, 6G, and S5B–S5H). We observed focal staining overlapping with *Xist* clusters for all antisera demonstrating that a repressive compartment and associated chromatin modifications can be established by *Xist* in Δ*Ubn1*, Δ*Ubn2*, Δ*Ubn1*/Δ*Ubn2*, and Δ*Hira* ESCs. The percentage of nuclei showing focal domains of H3K27me3 and H2AK119ub1 was significantly reduced in Δ*Ubn1*/Δ*Ubn2* ESCs (Fig 6D, 6F), whereas the reduction in focal JARID2 and

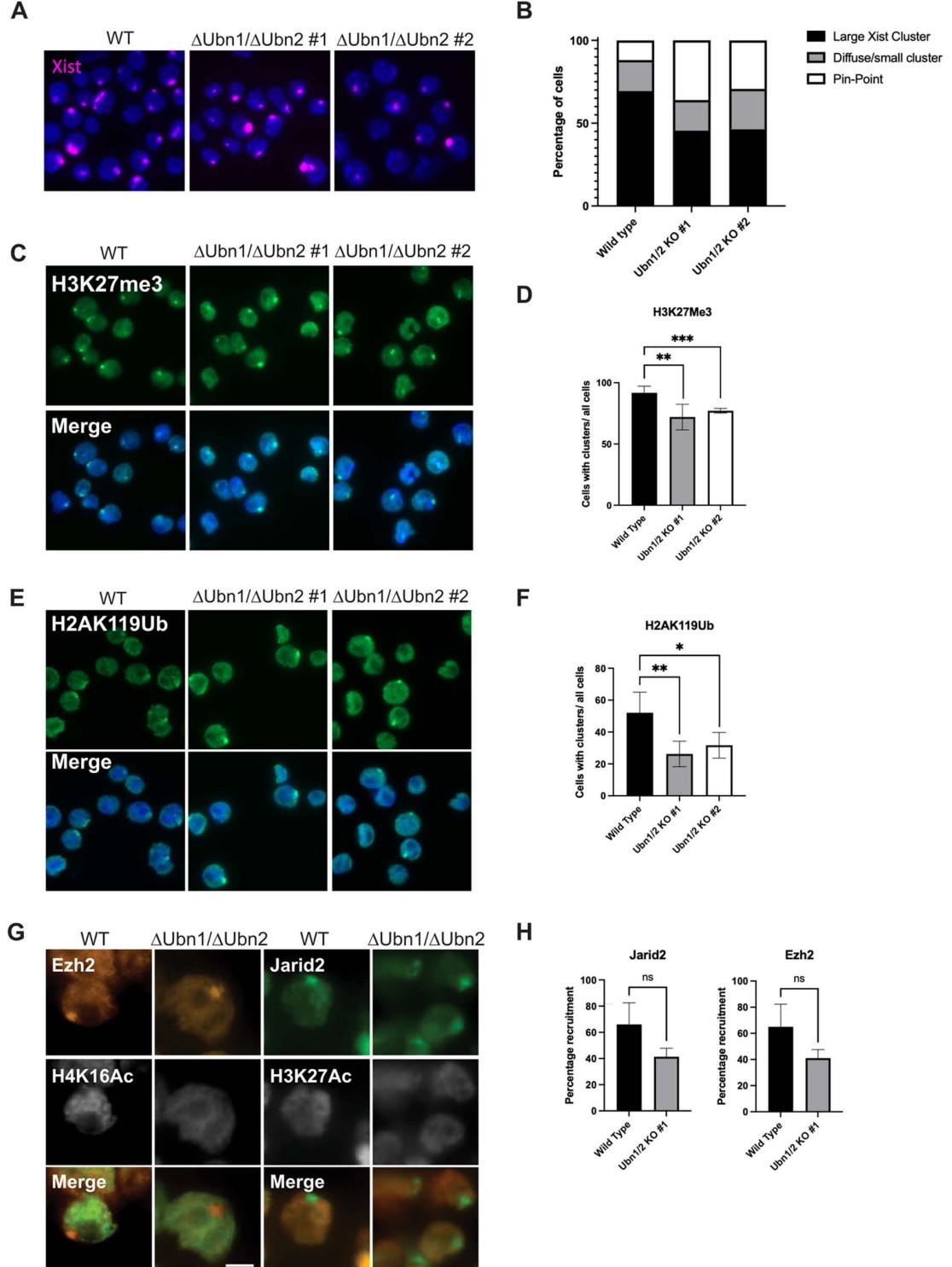

**Fig 6. Formation of a repressive compartment by *Xist* in *Ubinuclein*, and *Hira* mutant ESCs.** **(A)** *Xist* RNA FISH in WT and ΔUbn1/ΔUbn2 cells and **(B)** quantification of *Xist* RNA localization patterns after 24h of *Xist* induction (cells counted WT: 372, Ubn1/2 KO #1: 402, Ubn1/2 KO #2: 426). **(C)** Immunofluorescence staining for H3K27me3 in WT and ΔUbn1/ΔUbn2 and **(D)** quantification of foci formation after 24h of *Xist* induction (cells counted WT: 180, Ubn1/2 KO #1: 155, Ubn1/2 KO #2: 294). **(E)** Immunofluorescence staining for H2AK119Ub in WT and ΔUbn1/ΔUbn2 and **(F)** quantification of foci formation after 24h of *Xist* induction (cells counted WT: 221, Ubn1/2 KO #1: 118, Ubn1/2 KO #2: 189). **(G)** Double immunofluorescence staining

in ΔUbn1/ΔUbn2 double mutant ESCs after 24h of Xist induction using H4K16ac or H3K27ac with EZH2 and JARID2 antisera, respectively. Scalebar; 5 μm. **(H)** Quantification of JARID2 and EZH2 foci in ΔUbn1/ΔUbn2#1 ESCs after 24h of Xist induction. ns; P > 0.05, *; P ≤ 0.05, **; P ≤ 0.01, ***; P ≤ 0.001 (cells counted WT: 319, Ubn1/2 KO #1: 312, Ubn1/2 KO #2: 309).

Ezh2 remained below statistical significance (Fig 6H). Taken together our data show that a domain of repressed chromatin was established by Xist in the absence of Ubinucleins or Hira.

### Ubinucleins are required for switching H3K27ac to H3K27me3 over X-linked genes

Xist forms a repressive compartment over non-genic regions of the X chromosome that has been separated from gene repression [24,36]. Gene repression requires Xist repeat A and Spen dependent pathways [20,30,31]. Previous studies have shown that Ubinucleins and Hira mediate histone H3.3 deposition and establishment of H3K27me3 over promoters and regulatory regions in ESCs [10,16]. To investigate effects on X-linked genes at higher resolution, we performed native chromatin immunoprecipitation sequencing (nChIP-Seq) with an H3K27me3 specific antiserum. For this we analyzed three independent WT and ΔUbn1/ΔUbn2 ESC clones without or after 48 hours of Xist induction. Before Xist induction promoters of low and medium expressed genes had on average higher H3K27me3 signals than highly expressed genes in both WT and ΔUbn1/ΔUbn2 ESCs as expected (S6A–S6C Fig). After Xist induction H3K27me3 increased for all groups of X-linked promoters and gene bodies (Figs 7A, and S6A–S6C). However, the increase was substantially smaller in ΔUbn1/ΔUbn2 than WT ESCs, whereby the largest difference was seen for highly expressed genes (Figs 7A, and S6A–S6C). Plotting of fold-change of H3K27me3 against fold-change in gene expression showed that genes, whose repression by Xist is affected in ΔUbn1/ΔUbn2 ESCs, are also failing to acquire H3K27me3 at their TSS and gene body (S6B Fig). We observed a bimodal distribution of H3K27me3 around the transcription start site (TSS) of highly or moderately expressed X-linked genes, whereas low expressed genes showed a uniform increase (S6A, S6C Fig). To understand if H3K27me3 establishment over regulatory elements was also affected by mutations in Ubn1 and Ubn2, we analyzed 5kb intervals around the center of previously described elements. We observed modest increases in H3K27me3 after Xist induction over X-linked strong EZH2 sites [26] and CpG islands in WT ESCs that was slightly reduced in ΔUbn1/ΔUbn2 ESCs (S6D Fig). Over X-linked moderate EZH2 sites [26], X-linked Formaldehyde-Assisted Isolated Regulatory Elements (FAIRE), CTCF sites [37], predicted enhancers [38], and the transcription start sites (TSS) of RefSeq genes Xist induction increased H3K27me3 strongly in WT ESC but only weakly in ΔUbn1/ΔUbn2 ESCs (S6D Fig). Autosomal nChIP-Seq read coverage was similar for all ESC clones and independent of Xist induction and genotype (Figs 7A and S6C). We conclude that Xist induction strongly increased H3K27me3 specifically over X-linked genes and regulatory elements in WT ESCs but only weakly in ΔUbn1/ΔUbn2 ESCs.

To investigate whether persisting active chromatin modifications at promoters might prevent establishment of H3K27me3 at X-linked genes after Xist induction in ΔUbn1/ΔUbn2 ESCs, we performed CUT&RUN analysis using antibodies specific for H3K27ac and H3K4me3 (Fig 7B, 7C). Defined H3K27ac peaks were observed over the TSS of active genes in WT and ΔUbn1/ΔUbn2 ESCs as expected. After Xist induction H3K27ac was markedly reduced at X-linked but not autosomal genes in WT and ΔUbn1/ΔUbn2 ESCs demonstrating that deacetylation of H3K27ac occurs largely independent of Ubinucleins (Fig 7B). CUT&RUN analysis of H3K4me3 showed clear peaks over the TSS of active genes (Fig 7C). We observed a modest reduction after Xist induction in both WT and ΔUbn1/ΔUbn2 ESCs. Taken together our data show that removal of active marks over X-linked gene promoters is accomplished by Xist via Ubinuclein independent mechanisms.

We further performed directed ChIP for the X-linked genes Mecp2, Jarid1c, and Klf8 with antibodies specific for H3K27ac, H3K4me3, H3K27me3, and H2AK119ub. A small reduction of H3K4me3 and H3K27ac as well as an increase in H3K27me3 and H2Aub was evident in control ESCs after 48 hours of Xist induction (S6E Fig). In ΔUbn1/ΔUbn2 ESCs

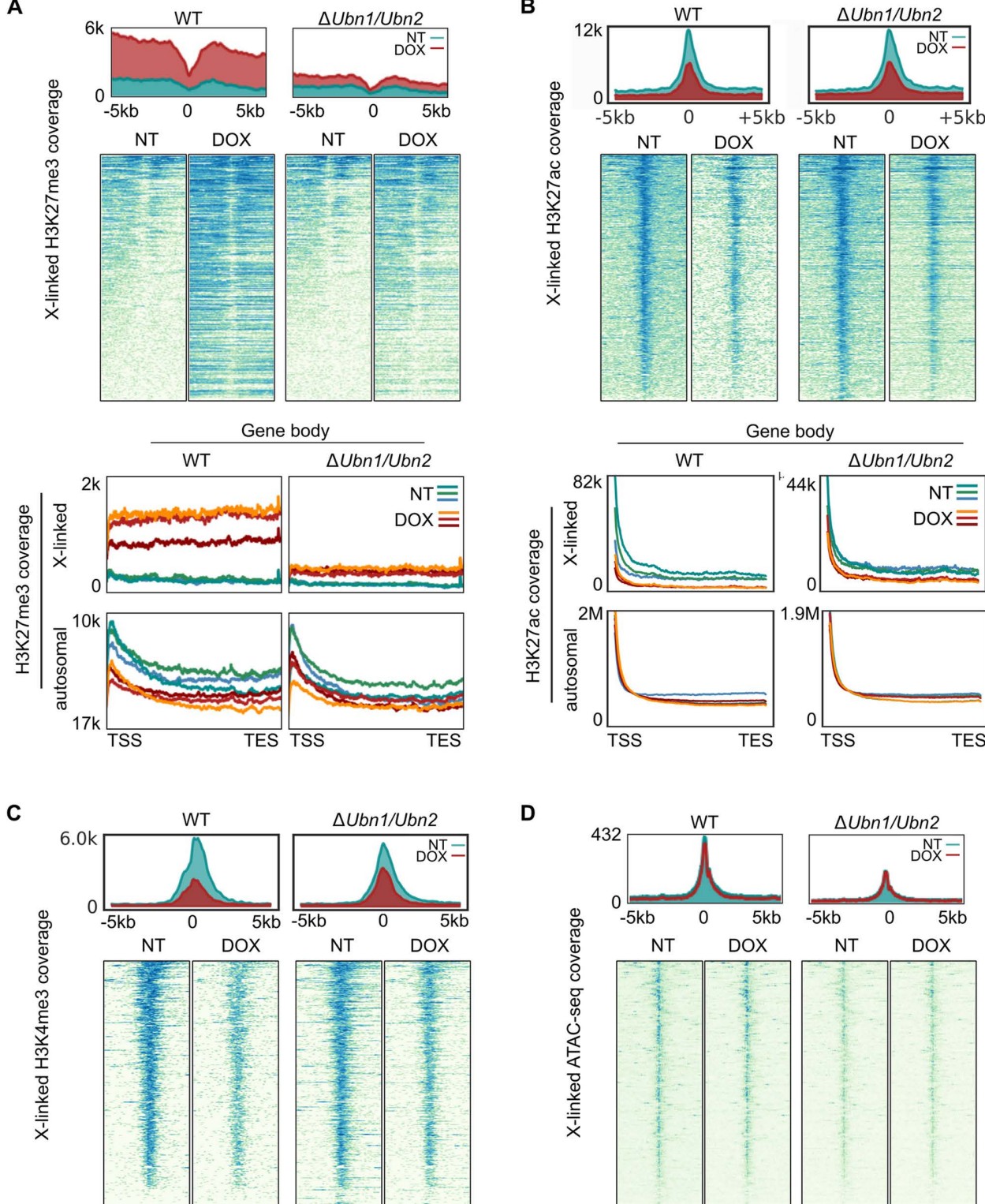

**Fig 7. Ubinucleins are required for H3K27ac to H3K27me3 switch during initiation of XCI.** Profiles and heatmaps showing cumulative (**A**) H3K27me3 ChIP coverage, (**B**) H3K27ac CUT&RUN signals, (**C**) H3K4me3 CUT&RUN coverage, and (**D**) ATACseq coverage over X-linked highly expressed genes for three independent replicates (**AB**) or a representative experiment (**CD**) of WT and ΔUbn1/ΔUbn2 ESC clones after 48h of *Xist*

induction (DOX) and without (NT). Profiles of 5kb around the TSS are shown. (**AB**) Profiles over the scaled transcription units of X-lined and autosomal highly expressed genes are shown for each of the three biological replicates (lower panels).

the reduction of H3K4me3 after *Xist* induction was variable. We also observed a markedly reduced increase in H3K27me3 after *Xist* induction compared to control ESCs. Taken together our findings demonstrate that loss of Ubinucleins causes a defect in establishment of H3K27me3 but deacetylation of H3K27ac over X-linked genes by *Xist* is unaffected.

### Initial repression by *Xist* has no effect on the nucleosome free region at the TSS of X-linked genes

Our observation of a bimodal distribution of H3K27me3 around the TSS of active genes indicated the presence of a nucleosome free region [9]. Notably, this gap became more pronounced at X-linked genes when H3K27me3 increased after *Xist* induction. To investigate chromatin accessibility directly, we performed Assay for Transposase-Accessible Chromatin using sequencing (ATACseq) in WT and Δ*Ubn1*/Δ*Ubn2* ESCs without and after 48 hours of *Xist* induction (Fig 7D). Induction of *Xist* slightly reduced overall accessibility over the X chromosome but not over autosomes in WT ESCs (S7A, S7B Fig) consistent with previous observations [39]. When the fold change in accessibility after *Xist* induction was plotted, a marginally smaller reduction was observed in Δ*Ubn1*/Δ*Ubn2* compared to WT ESCs (S7A Fig, lower panel). ATAC-Seq coverage over X-linked EZH2 stations, CpG islands, CTCF sites, and regulatory regions showed only minimal changes of accessibility after *Xist* induction in WT and Δ*Ubn1*/Δ*Ubn2* ESCs (S7C–S7E Fig).

ATAC-Seq profiles showed that the accessibility was similar over autosomes, but reduced over the X chromosome in Δ*Ubn1*/Δ*Ubn2* compared to WT ESCs before *Xist* induction. The reduced read coverage is likely explained by the loss of the second X chromosome in Δ*Ubn1*/Δ*Ubn2* ESCs (S3D Fig). *Xist* induction led to a moderate reduction of accessibility over the promoters and gene bodies of low and moderately expressed X-linked genes in WT and to a lesser extent in Δ*Ubn1*/Δ*Ubn2* ESCs (S7F Fig). Autosomal genes were unaffected by *Xist* induction as expected (S7G Fig). Notably, a peak of high accessibility directly at the TSS persisted after *Xist* induction at a time when gene repression became evident in our RNA-Seq analysis (Figs 7D, and S7F). Therefore, a nucleosome free region at the promoter of X-linked genes is maintained during the initiation of XCI. We conclude that impairment of *Xist* function in Δ*Ubn1*/Δ*Ubn2* ESCs is not explained by a requirement of HIRA complex mediated deposition of a nucleosome at the TSS of X-linked genes.

## Discussion

Our study characterizes the function of the Ubinuclein genes in mouse development. The finding that *Ubn1* is not essential in mice contrasts the earlier view of UBN1 as the principal Ubinuclein component of the HIRA complex [4]. We observe that homozygous *Ubn1* mutant mice are healthy and fertile. Our mutation deletes a large N-terminal region of UBN1 protein that has been shown to be required for interactions with HIRA and histone H3.3 [10,17,32]. Although the remaining exons encoding the Ubinuclein middle domain, which mediates dimerization, are present, Western analysis does not detect a predicted truncated protein in liver nuclei of *Ubn1* homozygous mice. In addition, combined mutations of *Ubn1* and *Ubn2* cause early embryonic lethality at a similar developmental stage as the *Hira* mutation. Taken together these observations strongly suggest that we have analyzed a loss-of-function mutation of *Ubn1*. The discrepancy to an earlier report finding *Ubn1* essential for development of mouse zygotes [33] can likely be reconciled by considering differences in genetic background or methods for gene inactivation. Furthermore, acute loss of *Ubn1* might have a more dramatic effect that might not resemble the constitutive mutation, and compensatory upregulation of *Ubn2* in *Ubn1* mutant embryos might be considered. However, we did not find compensatory regulation in *Ubn1* mutant ESCs. Indeed, none of the mutations in HIRA complex components that we have analyzed affected expression of the remaining *Hira*, *Ubn1*, or *Ubn2* genes. Our data support the view that, although *Ubn1* and *Ubn2* both contribute to mouse development, *Ubn2* alone can support all functions of the HIRA complex in mice.

## Phenotypic variation of the *Ubn2* mutation causes embryonic lethality dependent on sex

We observe striking variations of the phenotypes of *Ubn2* mutant embryos causing embryonic lethality with incomplete penetrance. Notably, the *Ubn2* mutation causes a dramatic deficit of females in C57BL/6 mice. This is likely due to a fortuitous genetic situation allowing males, albeit at a lower than expected rate, to survive. *Xist* and *Smchd1* have been the only genes associated with dosage compensation, whose mutation results in a female specific lethality [40,41]. Notably, preferential survival of males has also been observed due to anti-inflammatory activity of sex hormones in a study of mutations that impair DNA replication and lead to DNA damage [42]. McNairn et al. observe the first effect on sex ratios at E12.5. In contrast, a reduction of the number of female *Ubn1⁻ᐟ⁻ Ubn2⁻ᐟ⁻* double mutant embryos occurs before E10.5 with embryos arresting before turning and, importantly, before sex hormones are produced. Similarly, the two phenotypic classes of the *Hira* mutation also arise before E10.5 [14]. We show that the severe class of the *Hira* mutant phenotype correlates with female sex, whereas the large majority of males associate with the mild class. Taken together with our observation on impaired *Xist* function in *Ubn2*, and *Hira* mutant ESCs our findings strongly suggest that female specific effects of mutations in HIRA complex genes are not the result of secondary sexual characteristics but of defects in XCI.

## Mechanism of HIRA complex function in XCI

XCI is a multi-step process. Using inducible *Xist* expression in ESCs, we find that *Hira* and Ubinucleins are required for X-linked gene repression. Analysis of chromatin changes pinpoints which step in XCI requires HIRA complex function. Our data show that a repressive compartment is formed and *Xist* causes deacetylation of H3K27ac over promoters of X-linked genes in the absence of Ubinucleins. However, recruitment of H3K27me3 and H2AK119ub1 is impaired suggesting effects of loss of Ubinuclein genes on PRC1 and PRC2, respectively. A function of the HIRA complex in PRC2 recruitment has previously been observed at bivalent and developmentally regulated genes [16,43]. Although, gene silencing in XCI can occur without PRC2 function and in the absence of H3K27me3 [44,45], our observations indicate a defect in switching active X-linked gene promoters into a repressed configuration. Deacetylation of H3K27ac is likely mediated by HDACs independent of HIRA and Ubinucleins. One candidate is HDAC3 that is activated by SPEN following its recruitment by *Xist* repeat A [20,30,31]. We also find that a repressive compartment is formed in fibroblasts of female *Ubn2⁻ᐟ⁻* embryos. Considering that the repressive compartment is formed over genomic repeats, which are not targeted by the HIRA complex, this observation is consistent with the role of *Ubn1* and *Ubn2* for histone H3.3 deposition in genes and regulatory elements [10].

Our study further defines the role of the HIRA complex in gene repression by *Xist*. We observe a consistent peak of accessibility in our ATACseq and H3K27me3 ChIP analysis before and after *Xist* induction independent of mutations in Ubinuclein genes. Therefore, occupation of the nucleosome free region is not an early step in the mechanism of gene repression of *Xist* that requires HIRA complex function. This is consistent with our previous analysis of the repression of *Xist* by the antisense *Tsix* RNA [46]. The nucleosome free region at the *Xist* promoter becomes occupied only when *Tsix* mediated repression becomes irreversible. However, we do not rule out an effect on dynamic positioning of nucleosomes around the TSS. Notably, in ESCs a well positioned -1 nucleosome is enriched in histone H3.3 and shows fast dynamics [9], and histone H3.3 has been suggested as a mark for the nucleosome free region at the TSS [47]. Further studies of histone H3.3 dynamics will be required to address this aspect fully. Considering that histone H3.3 is not enriched over silent genes [48], the HIRA complex is likely involved in switching an active to a repressed chromatin configuration at X-linked genes. Dynamic turnover of histones might be expected to contribute to the removal of active chromatin modification. However, we find that deacetylation of H3K27ac is independent of HIRA complex function and removal of active marks is unaffected in Δ*Ubn1*/Δ*Ubn2* ESCs. An interesting possibility is that gene repression by *Xist* might depend on a specific feature of histone H3.3. Histone H3.3 but not H3.1 and H3.2 possesses a serine at position 31, which can be phosphorylated and has been implicated in gene regulation [49,50]. Therefore, it is enticing to speculate that serine 31 might play a role in the mechanism of gene repression by *Xist*.

A striking observation of our study is the partial loss of *Xist* function in the absence of *Hira* or Ubinucleins. This finding indicates the existence of parallel pathways at the initiation of XCI. One potential candidate is HDAC3. A recent study has observed that HDAC3 is already present on the X chromosome before *Xist* induction and a partial loss of gene repression is observed in ESCs with an *Hdac3* mutation [31]. Our observation that deacetylation of H3K27ac at X-linked gene promoters does not require Ubinucleins is consistent with a parallel pathway supporting the residual silencing function of *Xist*. Conceivably, HIRA complex function might also be partially compensated by other nucleosome assembly complexes. Histone H3.3 deposition is mediated by ATRX and DAXX at the centromeres and telomeres [51]. Notably, ATRX associates with the Xi in somatic and trophoblast stem cells [52]. However, *Atrx* has been implicated in the maintenance of XCI and not the initial establishment of gene repression by *Xist*. Our mutant ESC and mouse lines will facilitate investigation of potential redundancy between *Hira* or Ubinucleins, and *Atrx* or *Daxx* in future studies.

## Materials and methods

### Ethics statement

All animal experiments were approved by the Cantonal Veterinary office, Waltersbachstrasse 5, 8090 Zurich, Switzerland and carried out under the cantonal permit number ZH153/2021 and national permit number 34039 in accordance with Swiss federal and cantonal regulations.

### Culture and gene editing of ESCs

HATX3 and HATX8 ESCs were cultured as described previously [20]. Xist was induced by adding 1 µg/ml doxycycline. gRNAs were designed using the CHOPCHOP [53], GuideScan [54], and E-CRISPR [55] and cloned into pX458 (Addgene, #48138). 2µg of the vector was transfected into 300000 ESCs using Lipofectamin 2000 (ThermoFisher). GFP positive ESCs were sorted into 96-wells and genotyped by PCR and Sanger sequencing. An HA tag was inserted into *Ubn2*, using 1µg of ssDNA donor template (5'-tttctttctcagatggaggccaaagtaaaggggacactaagttaccacggaaacctcagTCCGCCTGGTCCCACCCTCAGTTTGAGAAAGGCGGCGGCTACCCCTACGACGTGCCCGACTACGCCTGActttccagcaaggggg agaggaaccacttggctggctggcgggaccgacctgatgggaag-3'). The *Hira* coding region was amplified from cDNA and cloned into a PiggyBac (Pb) vector containing an EF1-alpha promoter and a Ruby fluorescence reporter (System Biosciences, Cat No. PB531A-2). The *Ubn2* cDNA (Dharmacon, #MMM1013-202770327) was cloned into the PiggyBac vector described above. The ΔBLLF1 deletion was introduced by cutting the *Ubn2* expression vector with two Cas9 RNPs.

### Gene editing in mice

20 pmol Cas9 protein mixed with 40 pmol of sgRNA in injection buffer (8 mM TrisHCl pH 7.4, 0.1 mM EDTA) adjusted to a final volume of 12.5 µl was used for pronuclear microinjection of C57BL/6 zygotes. Mutations were identified by PCR using DNA from ear biopsies from founder mice.

### RNA isolation and RNA-Seq analysis

Total RNA was isolated using RNeasy Mini Kit (Quiagen, #74104) and DNA removal by on-column DNAse I digestion (Qiagen, #79254). 500 ng total RNA were used for cDNA synthesis using the PrimeScript RT Master Mix kit (Takara, #RR036A). Real-time quantitative PCR was performed using SYBER Green on a LightCycler 480 system (Roche). RNA-Seq was performed in triplicates for *Xist* inducible WT, Δ*Ubn1*, Δ*Ubn2*, Δ*Ubn1*/*Ubn2,* and Δ*Hira* ESCs non-treated and treated with Dox for 48h. Total RNA was isolated as above and used for NGS library preparation with the Illumina TruSeq Stranded mRNA kit. Sequencing was performed on a Novaseq 6000 (Illumina, Inc, California, USA) using single-end 100 bp sequencing.

## Co-Immunoprecipitation

2 mg of ESC lysate was incubated over night at 4°C on a rotator with 250U of Benzonase nuclease (Sigma-Aldrich, #E1014) and 55 µl (dry volume) of pre-washed anti-HA affinity matrix (Roche, #11815016001) in a final volume of 1 ml RIPA buffer. Subsequently, beads were washed twice for 5 min under rotation with wash buffer (50 mM TrisHCl pH 8, 150 mM NaCl, 1 mM EDTA, 0.02% NP40), and once with PBS. Proteins were eluted from the beads by resuspending in 2 x Laemmli buffer and boiling for 5 min at 95°C.

## Immunofluorescence and RNA FISH analysis

For immunofluorescence staining, cells were grown on multiwell Roboz Slides (Cellpoint Scientific, USA), and then fixed and permeabilized. Primary antibodies were diluted in blocking buffer and incubated over night at 4°C followed by incubation with secondary antibodies for 1h at RT. Nuclei were counterstained with 1.4 µM DAPI (Molecular Probes, #D1306). After several washes in PBS/ 0.1% Tween-20 (PBST) slides were mounted using Vectashield (Reactolab, #H1000), or Mowiol (Sigma-Aldrich, #81381). For combined immunofluorescence RNA FISH analysis, incubation with primary and secondary antibody was 45 min at room temperature using RiboLock RNAse inhibitor (Thermo Scientific, #EO0381). Slides were postfixed, air dried, and hybridized with Cy3-labeled *Xist* RNA FISH probe (Cy3 dCTP, GE Healthcare Amersham) generated by random priming with Prime-it II, (Stratagen, #300385) and competed with mouse Cot-1 DNA (Invitrogen, #18440-016), salmon sperm DNA (Invitrogen, #15632-011) and tRNA (Invitrogen, #AM7119) in Hybrisol VII (MP Biomedicals). After overnight incubation, and washing, nuclei were counterstained with a 1.4 µM DAPI solution (Molecular Probes, #D1306).

## Native ChIP and ChIP-Seq analysis

All buffers contained Complete Mini protease inhibitor (Roche, #11836153001) and 5 mM Na-butyrate deacetylase inhibitor. 12 million ESCs were pelleted and resuspended in 90 µl of Lysis Buffer (LB) (50 mM TrisHCl pH 7.5, 150 mM NaCl, 0.1% sodium deoxycholate, 1% Triton, 5 mM CaCl2). Chromatin was digested using 60 µl of LB containing 0.3 µl Micrococcal nuclease (Cell signalling, #10011S) at 37°C for 15 min. Reactions were stopped with 15 µl 0.5 M EDTA, and Stop Buffer (SB) (50 mM TrisHCl pH 7.5, 150 mM NaCl, 0.1% sodium deoxycholate, 1% Triton, 30 mM EGTA, 30 mM EDTA). After adjusting the volume to 1 ml with LB/ SB (1:1), 30 µl were used as input. 300 µl of chromatin was adjusted to 1ml with LB/ SB (1:1) and incubated over night with corresponding antibody at 4°C. 15 µl Dynabeads protein G (ThermoFisher, #10004D) blocked with blocking buffer (PBS 1X, 0.5% Tween, 0.5% BSA) were added and incubated 2h at 4°C. After washing, DNA was eluted from beads and purified using the MinElute PCR purification kit (Qiagen, #28004). ChIP-Seq was performed in triplicates for *Xist* inducible WT and Δ*Ubn1*/*Ubn2* ESCs, witout or after 48h of *Xist* induction. Chromatin was prepared as above with digestion with Micrococcal nuclease for 20 min. 1 ng DNA was used for library preparation using the NEBNext ultra II DNA Library Prep Kit from NEB (New England Biolabs (NEB), Ipswich, MA, #7103). Libraries were size-selected for fragments of 300–600 bp with Agencourt AMPure XP magnetic beads (Beckman Coulter), and sequenced on an Illumina HiSeq 2500 machine using 125 bp single-end sequencing.

## CUT&RUN analysis

CUT&RUN was performed using CUT&RUN Assay Kit (Cell SignalingTechnology #86652) and antibodies against H3K27ac (Cell Signaling Technology #8173, 1:50) and H3K4me3 (Cell Signaling Technology rabbit mAb #9751, 1:50). DNA was purified from enriched chromatin samples using Spin Columns (Cell Signaling Technology #14209). For library prep, NEBNext Ultra II DNA Library Prep Kit for Illumina (NEB #E7103) was used. Libraries were sequenced on an Illumina MiSeq or Illumina NextSeq 2000 instrument.

PLOS Genetics

## ATACseq

ATACseq was performed in triplicates following the improved version of the original protocol [56]. ESCs were lysed, and nuclei incubated with TDE1 tagment DNA enzyme (Illumina, #15027865) in tagment DNA buffer (Illumina, #15027866). DNA was purified with MinElute PCR purification kit, amplified and multiplexed with Nextera DNA CD indexes (Illumina, #20015881). Libraries were purified with MinElute PCR purification kit and used for paired-end 75 bp sequencing on a HS 2500 illumina instrument.

## Computational analysis of next generation sequencing data

For the RNAseq analysis, Trimmomatic [57] and HISAT2 [58] were used for pre-processing of the reads including trimming of adaptors and alignment of reads to the mm10 mouse genome reference assembly, respectively. HTSeq was used for counting RNAseq reads that overlap gene features, handling of genome annotation, and performing data analysis involving genomic coordinates [59]. Data was normalized to transcripts per million (TPM) for all analyses except for differential gene expression. DESeq2version 1.38.3 was used for the analysis of differential gene expression and calculation of adjusted p-values from raw read counts of RNAseq datasets [35]. For the analysis of ChIPseq, ATACseq, and CUT and RUN datasets Trimmomatic [57] and Bowtie2 [60] were used for pre-processing of the reads including trimming of adaptors and alignment of reads to the mm10 mouse genome reference assembly, respectively. HTSeq was used for counting reads that overlap gene features, handling of genome annotation, and performing data analysis involving genomic coordinates [59]. Python and R scripts, genomic annotation, and intermediate data for computational analyses are made available on github at https://github.com/WutzLab.

## Statistics

Paired, parametric, two-tailed t-tests were performed to determine significance with a p-value threshold smaller than 0.05. Calculations were performed using Prizm Graphpad Version 7.

## Supporting information

**S1 Text. Includes the Supplementary methods, S1 Table legend, S2 Table legend, S4Table legend, and Supplementary References.**
(PDF)

**S1 Fig. Contains S1 Fig and legend.**
(PDF)

**S2 Fig. Contains S2 Fig and legend.**
(PDF)

**S3 Fig. Contains S3 Fig and legend.**
(PDF)

**S4 Fig. Contains S4 Fig and legend.**
(PDF)

**S5 Fig. Contains S5 Fig and legend.**
(PDF)

**S6 Fig. Contains S6 Fig and legend.**
(PDF)

**S7 Fig.** Contains S7 Fig and legend.
(PDF)

**S1 Table.** Includes S1 Table showing numbers and statistics of *Ubn2* mutant mouse crosses.
(XLSX)

**S2 Table.** Includes S2 Table showing numbers and statistics of *Ubn1* and *Ubn2* double mutant mouse crosses.
(XLSX)

**S3 Table.** Includes S3 Table showing SILAC mass spectrometry data of HA-Ubn2 interactors.
(PDF)

**S4 Table.** Includes S4 Table as an archive of 4 Excel workbooks with lists of differentially regulated genes in ΔUbn1, ΔUbn2, ΔUbn1/ΔUbn2, and ΔHira mutant compared to WT ESCs.
(ZIP)

**S5 Table.** Includes S5 Table showing sequences of primers and gRNAs used in this study.
(PDF)

**S6 Table.** Includes S6 Table showing a list of reagents, instruments, and materials used in this study.
(PDF)

**S1 Data.** Contains source data for graphs in Fig 4.
(XLSX)

**S2 Data.** Contains source data for Western blots in Fig 4.
(PDF)

**S3 Data.** Contains source data for graphs in Fig 5.
(XLSX)

**S4 Data.** Contains source data for graphs in Fig 6.
(XLSX)

## Acknowledgments

We thank R. Freimann and T. Henneck for kindly providing flow cytometry and transgenics services, and A. Postlmayr for discussion. PJS thanks the British Heart Foundation for support.

## Author contributions

**Conceptualization:** Asun Monfort, Giulio Di Minin, Sarah Sting, Charles Etienne Dumeau, Peter Scambler, Anton Wutz.

**Data curation:** Asun Monfort, Giulio Di Minin, Sarah Sting, Anton Wutz.

**Formal analysis:** Asun Monfort, Giulio Di Minin, Sarah Sting, Anton Wutz.

**Funding acquisition:** Anton Wutz.

**Investigation:** Asun Monfort, Giulio Di Minin, Sarah Sting, Charles Etienne Dumeau, Anton Wutz.

**Methodology:** Asun Monfort, Giulio Di Minin, Sarah Sting, Charles Etienne Dumeau.

**Project administration:** Asun Monfort, Anton Wutz.

**Resources:** Giulio Di Minin, Charles Etienne Dumeau, Peter Scambler.

**Software:** Asun Monfort, Giulio Di Minin, Sarah Sting, Anton Wutz.

**Supervision:** Asun Monfort, Anton Wutz.

**Validation:** Asun Monfort, Sarah Sting.

**Visualization:** Asun Monfort, Giulio Di Minin, Sarah Sting, Anton Wutz.

**Writing – original draft:** Asun Monfort, Peter Scambler, Anton Wutz.

**Writing – review & editing:** Asun Monfort, Giulio Di Minin, Sarah Sting, Charles Etienne Dumeau, Peter Scambler, Anton Wutz.

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
