## [Decision Letter · Decision Letter 0]

18 Sep 2024

Dear Dr Wutz,

Thank you very much for submitting your Research Article entitled 'Ubinuclein 2 is essential for mouse development and functions in X chromosome inactivation' to PLOS Genetics.

The manuscript was fully evaluated at the editorial level and by independent peer reviewers. The reviewers appreciated the attention to an important problem, but raised some substantial concerns about the current manuscript. Based on the reviews, we will not be able to accept this version of the manuscript, but we would be willing to review a much-revised version. We cannot, of course, promise publication at that time.

If you decide to revise the manuscript for further consideration at PLOS Genetics, please aim to resubmit within the next 60 days, unless it will take extra time to address the concerns of the reviewers, in which case we would appreciate an expected resubmission date by email to plosgenetics@plos.org.

If present, accompanying reviewer attachments are included with this email; please notify the journal office if any appear to be missing. They will also be available for download from the link below. You can use this link to log into the system when you are ready to submit a revised version, having first consulted our Submission Checklist .

PLOS has incorporated Similarity Check , powered by iThenticate, into its journal-wide submission system in order to screen submitted content for originality before publication. Each PLOS journal undertakes screening on a proportion of submitted articles. You will be contacted if needed following the screening process.

To resubmit, log into your Editorial Manager account and select the option 'Revise Submission' in the 'Submissions Needing Revision' folder.

We are sorry that we cannot be more positive about your manuscript at this stage. Please do not hesitate to contact us if you have any concerns or questions.

Yours sincerely,

Christine Wells

Academic Editor

PLOS Genetics

Aleksandra Trifunovic

Section Editor

PLOS Genetics

Reviewer's Responses to Questions

**Comments to the Authors:**

Reviewer #1: In this manuscript Monfort and colleagues study the HIRA complex cofactors Ubn1 and Ubn2, finding that mutation of Ubn2 but not Ubn1 causes partially penetrant embryonic lethality. Ubn1/2 double mutation causes fully penetrant embryonic lethality, with increased developmental arrest in females. The authors find that Ubn1/2 and Hira leads to impaired gene silencing by XCI and H3K27me3 deposition.

These findings are of very high interest to the XCI community and will warrant publication in PLoS genetics. The implication, although not explored here, that H3.3 may play a role in XCI is particularly enticing. However, many of the experiments presented are not fully quantified with appropriate statistical analysis, which will need to be performed.

Major comments

1. More quantitation of the data should be performed, together with appropriate statistical testing for data where conclusions are drawn, particularly for colony counting and IF experiments. The claim that XCI failure is greater in the Ubn1/2 dko than the single ko’s appears valid by eye from the RNAseq, but this should also be tested statistically.

2. The Fig5 legend says that a p-value cut off was used for DE genes in the RNAseq. This would be inappropriate as it implies no correction for multiple testing. Please clarify whether multiple testing was performed, and if not, present only multiple testing corrected results.

3. What is the explanation for there being no overlap in DE genes between the different mutants? This would seem to contradict the claim that these proteins are in complex together. Genomic experiments are performed 48 hours after Xist induction, however it is not clear when the cells begin to die. Is it possible that the RNAseq readout is clouded by cells in the process of dying?

4. The text says that Xist foci were the same between mutants, however the FISH data in Fig6a appears to show greatly reduced Xist localisation in the mutants. Have I misinterpreted something? If not, I think foci volume should be quantified.

Minor comments

1. It wasn’t clear to me if the HATX cells were female, male or haploid. Please clarify.

2. The ordering references to Fig4 were incorrect in the text.

3. There is a typo in this sentence – “Xist forms a repressive compartment over non-genic regions of the X chromosome.” It should read “genic” regions.

Reviewer #2: The review is uploaded as an attachment.

Reviewer #3: Comments to the authors

In this manuscript, Monfort et al. investigate role of the HIRA complex including UBN1/UBN2 in X chromosome inactivation (XCI). They demonstrated that KO mice of Ubn1/Ubn2 or Hira have severe female-specific embryonic lethality, suggesting their important roles in XCI in vivo. They also showed that the mutation of Ubn1/Ubn2 or Hira in mouse ESCs compromises Xist-mediated gene silencing on the inactive X chromosome (Xi). Mechanistically, H3K27me3 accumulation on the Xi upon Xist induction is impaired in Ubn1/Ubn2 or Hira mutant ESCs.

I think the in vivo KO mouse data is straightforward and convincing for their conclusions. My concern is, however, there are several points still preliminary or unclear for me and these points should be clearly answered to improve this study.

Major concerns

1. Although the authors described “The percentage of nuclei showing focal domains of JARID2, and H3K27me3 overlapping with Xist was slightly lower in ΔUbn2 and ΔHira than control ESCs after 24 hours of Xist induction, whereas H2AK119ub1 remained unchanged (Fig. 6AC and Suppl. Fig. 5B-E).” on page 12, I can’t seem to find the images of H2AK119ub1 staining. As the Brockdorff lab previously showed the presence of Xist-PRC1-PRC2 cascade in XCI initiation (Almeida et al 2017), I think these staining images of H3K27me3 and H2AK119ub1 are important and should be displayed in a main figure. In addition, several images such as RING1B staining in Fig 6 and Suppl Fig 5 lack quantification. It is necessary to show quantification charts for all of the imaging data with statistic evaluation. The number of counted cells should also be indicated.

2. Relating to 1, H3K27me3 ChIP-seq was performed with ∆Ubn1/∆Ubn2 ESCs (Fig 7), but H3K27me3 immunostaining was performed with ∆Ubn2 and ∆Hira ESCs (Suppl. Fig. 5BCDE), not with ∆Ubn1/∆Ubn2 ESCs. Is it possible to show an image and quantification data about H3K27me3 in ∆Ubn1/∆Ubn2 ESCs? I think this data is important to support the idea that overall H3K27me3 accumulation (IF data) is only partially impaired by the HIRA complex mutations but local H3K27me3 accumulation (ChIP-seq) is strongly attenuated in these mutant ESCs by comparing IF and ChIP-seq in the same genetic background.

3. By using ChIP-seq analysis, the authors showed that H3K27me3 accumulation upon Xist induction is impaired in ∆Ubn1/∆Ubn2 or ∆Hira ESCs at around TSS and gene body (Figure 7A). However, immunostaining of EZH2/H3K27me3 indicated the accumulation of PRC2/H3K27me3 on the Xi is only partially impaired in ∆Ubn2 and ∆Hira ESCs (Suppl. Fig. 5BCDE). These data suggest that Xi chromatin change in the HIRA complex mutants is not X chromosome-wide. Rather, it might be limited to certain genomic regions on the Xi. My concern is that the HIRA complex mutants would have a lack of H3.3 incorporation at limited regions such as around TSS and gene body on the Xi, which affects H3K27me3 accumulation on the Xi partially. Therefore, I think it is important to show whether H3.3 is actually incorporated into the Xi chromatin upon Xist induction in initiation of XCI and how much of them is inhibited in the HIRA complex mutant ESCs. This experiment will clarify the molecular mechanism of HIRA-dependent XCI regulation. Is it possible to examine H3.3 ChIP-seq and immunostaining with an anti-H3.3 antibody in Xist-inducible ESCs?

4. A sentence “ATAC-Seq profiles showed that accessibility was similar in WT and ΔUbn1/ΔUbn2 ESCs before Xist induction.” on page 15 would be misleading as the X chromosome-wide ATAC-seq data shown in Supple. Fig, 7A appears a clear decrease of ATAC-seq peaks in ∆Ubn1/∆Ubn2 ESCs even before Xist induction.

5. They conclude that insufficient H3K27me3 accumulation by Xist in HIRA complex mutant ESCs might be a cause of impaired silencing of X-linked genes in these ESCs. As they have H3K27me3 ChIP-seq data and RNA-seq data, is it possible to compare the degree of insufficient H3K27me3 accumulation and the level of impaired gene silencing for X-linked genes in the mutants? I think this comparison could further strengthen the role of H3K27me3 deposition in Xist-mediated gene silencing via Xist-HIRA axis.

Minor concerns

1. Indication of genotypes in each figure should be in the same way. For example, some figures use ∆Ubn1/Ubn2 and the other figures use ∆Ubn1/∆Ubn2.

2. Similarly, Y-axis title is lacking in many charts. For example, all ChIP-seq data lack Y-axis title and need to be correctly indicated.

3. Page 5

Edith Heard lab showed that only B repeat mutant is not sufficient to completely block the PRC1/2 accumulation on the Xi. Rather, they showed that both B and C repeats of Xist RNA is required for PRC1/2 recruitment on the Xi (Ref 29). This should be correctly described.

4. Figure 4CFM, Suppl Figures 2F&3AB

The charts in these figures present the result of colony forming assays with various ESC clones but are a bit confusing because two different forms of charts are used there. Is it possible to use the same form of chart in these figures?

5. Suppl Figure 4AB

To help to recognize which genomic region is deleted in these clones, would it be possible to show the targeted region on the top of each map?

6. Suppl Figure 5A

Images are too small to see the distribution of HIRA and UBN2. Enlarged images (ideally 2 or 3 cells per image) should be presented here.

7. The indication “Ubn2-HA ∆Ubn1/∆Ubn2” is confusing and one can misunderstand these cells express exogeneous Ubn2-HA transgene. Can another name be used for this cell line? I think probably simply “∆Ubn1/∆Ubn2” or “∆Ubn1/∆Ubn2-HA” would be fine.

8. Pages 9 & 14

There are several typos. Fig. 1G on the page 9 and H3K4me4 on the page 14.

**Have all data underlying the figures and results presented in the manuscript been provided?**

Reviewer #1: Yes

Reviewer #2: Yes

Reviewer #3: Yes

PLOS authors have the option to publish the peer review history of their article (what does this mean? ). If published, this will include your full peer review and any attached files.

**Do you want your identity to be public for this peer review?** For information about this choice, including consent withdrawal, please see our Privacy Policy .

Reviewer #1: No

Reviewer #2: No

Reviewer #3: No

---

## [Editor Report · Decision Letter 1]

27 Jan 2025

PGENETICS-D-24-00854R1

Ubinuclein 2 is essential for mouse development and functions in X chromosome inactivation

PLOS Genetics

Dear Dr. Wutz,

Thank you for submitting your manuscript to PLOS Genetics. After careful consideration, we feel that it has merit but does not fully meet PLOS Genetics's publication criteria as it currently stands. Therefore, we invite you to submit a revised version of the manuscript that addresses the points raised during the review process.

Please submit your revised manuscript within 60 days Mar 28 2025 11:59PM. If you will need more time than this to complete your revisions, please reply to this message or contact the journal office at plosgenetics@plos.org. Please include the following items when submitting your revised manuscript:

We look forward to receiving your revised manuscript.

Kind regards,

Christine Wells

Associated Editor

PLOS Genetics

Aleksandra Trifunovic

Section Editor

PLOS Genetics

Aimée Dudley

Editor-in-Chief

PLOS Genetics

Anne Goriely

Editor-in-Chief

PLOS Genetics

**Journal Requirements:**

We noticed that you used the phrase 'data not shown' in the manuscript. We do not allow these references, as the PLOS data access policy requires that all data be either published with the manuscript or made available in a publicly accessible database. Please amend the supplementary material to include the referenced data or remove the references.

**Reviewers' comments:**

**Figure resubmission:**
---

## [Decision Letter · Decision Letter 2]

4 May 2025

Dear Dr Wutz,

We are pleased to inform you that your manuscript entitled "Ubinuclein 2 is essential for mouse development and functions in X chromosome inactivation" has been editorially accepted for publication in PLOS Genetics. Congratulations!

Before your submission can be formally accepted and sent to production you will need to complete our formatting changes, which you will receive in a follow up email. In addition, we ask that you complete any minor changes requested by the reviewers at this time. Please be aware that it may take several days for you to receive your formatting email; during this time no action is required by you. Please note: the accept date on your published article will reflect the date of this provisional acceptance, but your manuscript will not be scheduled for publication until the required changes have been made.

Yours sincerely,

Paula E. Cohen

Section Editor

PLOS Genetics

Aleksandra Trifunovic

%CORR_ED_EDITOR_ROLE%

PLOS Genetics

Aimée Dudley

Editor-in-Chief

PLOS Genetics

Anne Goriely

Editor-in-Chief

PLOS Genetics

Comments from the reviewers (if applicable):

Reviewer's Responses to Questions

**Comments to the Authors:**

Reviewer #1: My initial comments and concerns have all been answered and I am happy with the manuscript. Congratulations to the authors on a nice study.

Reviewer #2: The revision has well addressed my comments.

Reviewer #3: Comments to the authors

I thank the authors for responding to all of my concerns. I think the revised manuscript becomes much better and will be appreciated by many readers of PLOS Genetics.

I still have a few minor concerns listed below about newly added dataset.

1. Figure 6E

Thank you for adding this new data. The new data shows H2AK119Ub accumulation on the Xi is also attenuated in Ubn1/2 KO ESCs, suggesting not only PRC2/H3K27me3 but also PRC1/H2AK119Ub may be a downstream of Ubn1/2. As PCGF3/5-PRC1/H2AK119ub has been reported to induce PRC2.2/H3K27me3 on the Xi, I think the authors should note a potential involvement of PRC1 in Ubn1/2’s function in the text.

2. Figure 6

How many cells were counted in each IF experiment? Number of cells counted in these experiments should be presented in the figure or legend.

3. Supplementary Figure 5B-E

I thank the authors for adding quantification of IF experiments. However, several IF experiments still do not have quantification such as RING1B in ∆Ubn2#2 and ∆Hira#3, and H2AK119ub in ∆Hira#3, in Suppl figure 5B-E. Relating to 1, quantification of RING1B staining is important to speculate the involvement of PRC1 in Ubc/Hira pathway. Is it possible to add quantification charts for them?

4. Supplementary Figure 6B

The current X-axis of lower panels is "fold change X-linked gene expression (+Dox/-Dox)”. I wonder this should be “Fold change of gene expression (+Dox/-Dox)”? Please confirm this point.

**Have all data underlying the figures and results presented in the manuscript been provided?**

Reviewer #1: None

Reviewer #2: Yes

Reviewer #3: Yes

PLOS authors have the option to publish the peer review history of their article (what does this mean? ). If published, this will include your full peer review and any attached files.

**Do you want your identity to be public for this peer review?** For information about this choice, including consent withdrawal, please see our Privacy Policy .

Reviewer #1: No

Reviewer #2: No

Reviewer #3: No

**Data Deposition**

http://datadryad.org/submit?journalID=pgenetics&manu=PGENETICS-D-24-00854R2

**Press Queries**

---

## [Editor Report · Acceptance letter]

PGENETICS-D-24-00854R2

Ubinuclein 2 is essential for mouse development and functions in X chromosome inactivation

Dear Dr Wutz,

We are pleased to inform you that your manuscript entitled "Ubinuclein 2 is essential for mouse development and functions in X chromosome inactivation" has been formally accepted for publication in PLOS Genetics! Your manuscript is now with our production department and you will be notified of the publication date in due course.

With kind regards,

Judit Kozma

PLOS Genetics

On behalf of:
